# Development of a Microwell System for Reproducible Formation of Homogeneous Cell Spheroids

**DOI:** 10.3390/pharmaceutics18010056

**Published:** 2025-12-31

**Authors:** Miguel A. Reina Mahecha, Ginevra Mariani, Pauline E. M. van Schaik, Paulien Schaafsma, Theo G. van Kooten, Prashant K. Sharma, Inge S. Zuhorn

**Affiliations:** 1Department of Biomaterials and Biomedical Technology, University Medical Center Groningen, University of Groningen, 9713 AV Groningen, The Netherlandsg.mariani@umcg.nl (G.M.); p.schaafsma@umcg.nl (P.S.); t.g.van.kooten@umcg.nl (T.G.v.K.); p.k.sharma@umcg.nl (P.K.S.); 2Molecular Neurobiology, Department of Biomedical Sciences of Cells and Systems, University Medical Center Groningen, University of Groningen, 9713 AV Groningen, The Netherlands

**Keywords:** vat photopolymerization, microwells, 3D cell culture, cell spheroids, pluripotent stem cells, embryoid bodies, tumor spheroids

## Abstract

**Background/Objectives:** Three-dimensional (3D) cell cultures are increasingly used because 3D cell aggregates better mimic tissue-level biological mechanisms and support studies of tissue physiology and drug screening. However, existing laboratory methods and commercial microwell platforms often yield inconsistent results and can be error-prone, time-consuming, or costly. The objective of this work was to develop a reproducible, high-yield, and cost-effective approach for generating homogeneous cell aggregates using custom 3D-printed microwell stamps. **Methods:** Custom conical and semi-spherical microwell stamps were fabricated using 3D printing. Stamp resolution was characterized by scanning electron microscopy (SEM). Negative imprints were cast in polydimethylsiloxane (PDMS), a biocompatible and hydrophobic polymer conducive to cell aggregation. These PDMS microwells were then used to generate pluripotent stem cell aggregates (embryoid bodies, EBs) and tumor spheroids from adherent cancer cell lines. **Results:** The 3D-printed stamps produced high-resolution conical and semi-spherical microwells in PDMS. Semi-spherical microwells enabled rapid, simple, and cost-effective formation of pluripotent stem cell aggregates that were homogeneous in size and shape. These aggregates outperformed those produced using commercial microwell plates and ultra-low attachment plates. The fabricated microwells also generated uniform tumor spheroids from adherent cancer cells, demonstrating their versatility. **Conclusions:** The in-house 3D-printed microwell stamps offer a reproducible, efficient, and economical platform for producing homogeneous cell aggregates. This system improves upon commercial alternatives and supports a broad range of applications, including pluripotent stem cell embryoid body formation and tumor spheroid generation.

## 1. Introduction

In the biomedical field, 3D printing is used to develop microfluidic lab-on-chip devices for tissue culture and drug testing [1,2]. Polydimethylsiloxane (PDMS), a low-cost, chemically inert polymer for rapid prototyping, is commonly used for these applications. This polymer has become a relevant substrate for biological analysis due to its biocompatibility, hydrophobicity, and stiffness tuneability [3]. Furthermore, PDMS shows optical transparency, gas permeability, and the possibility to be molded into any geometrical shape [4]. Profiting from these properties, PDMS has been used as a substrate to form cell aggregates [5].

Three-dimensional cell aggregates better mimic the in vivo environment of tissues and organs than conventional 2D cell cultures, including the presence of nutrient and oxygen gradients [6,7]. Thus, 3D cell cultures enhance the relevance of in vitro cell models for studying biological processes, disease mechanisms, and drug responses [8]. Three-dimensional cell aggregates or spheroids can be produced from multiple cell types, including stem cells, tissue-specific cells, and cancer cells. Spheroids derived from pluripotent stem cells, including embryonic stem cells (ESCs) and induced pluripotent stem cells (iPSCs), are called embryoid bodies (EBs) and can be induced to differentiate into different cell lineages [9]. Although the formation of EBs is inherent to all pluripotent stem cells, research largely focuses on using iPSCs due to the ethical issues linked to using ESCs.

iPSCs are reprogrammed somatic cells that have the ability to differentiate into all cell types in the body [10]. iPSC technology has opened the door for creating patient-specific and disease-specific cell models to develop personalized and disease-specific treatments and technologies, while simultaneously overcoming the ethical limitations of using ESCs. Furthermore, for human disease modeling, patient-specific iPSCs present advantages over genetically modified ESCs, because intrinsically they have the genetic and epigenetic profiles from the patient they were derived from [11]. However, although iPSCs and ESCs are functionally similar, they have been shown to be biochemically diverse, specifically in nucleic acid content [12], which warrants further investigation.

Different systems have been developed and tested for the formation of cell aggregates in order to investigate early embryogenesis and tumor formation [6]. The hanging drop method is one of the most common methods to produce EBs and tumor spheroids [13]. With this technique, multiple cell aggregates are typically formed within a single drop, and their sizes and shapes are inconsistent [14]. In addition, the feeder medium cannot be easily replaced, limiting drug testing and differentiation screening within the aggregates. An alternative method for creating cell aggregates is through the use of ultra-low-adherent (ULA) cell culture plates, which also typically results in the formation of cell aggregates with different shapes and sizes. Because size and shape determine nutrient and oxygen availability in the cell aggregates, consistent geometry of the cell aggregates is important for obtaining reliable results in subsequent experiments. For example, EB geometry has been shown to affect their differentiation potential [9]. Lastly, centrifugation of cells in round-bottom ULA plates or Eppendorf tubes covered with agarose gives single cell aggregates with similar sizes but heterogeneous shapes, with high variability and low reproducibility [6]. With this method, the cell aggregates need to be treated one by one in subsequent experiments, which is laborious and susceptible to human error.

Micromolds have been studied as novel aggregation platforms to maximize cell aggregate production and minimize heterogeneity, although the homogeneity of cell aggregate size and shape remains an issue [15]. Therefore, this paper focuses on employing vat polymerization (VP) 3D printing for rapid prototyping of microwell designs for cell aggregate fabrication with increased homogeneity and controllable size and shape. Three-dimensional (3D) printing technology, i.e., the layer-by-layer fabrication of 3D geometrical structures based on a digital model created by computer-aided design (CAD) software, has found applications in many fields, including healthcare [16,17]. Of the multiple methods of 3D printing [18], VP remains advantageous due to its possibility of constructing free-standing microarchitectures with outstanding resolution and accuracy [19] (Appendix A). Here, VP 3D-printed stamps were made of a commercially available biocompatible resin. The stamps were used to imprint (truncated) conical and semi-spherical microstructures into PDMS to create inserts for 6-well plates (Figure 1). The newly developed PDMS inserts were compared with the Aggrewell™ 800 and the Primesurface^®^ commercial plates for EB formation from hESCs and iPSCs. The semi-spherical and conical microwell configurations were optimized for cell aggregate fabrication with high homogeneity and ease of handling. The PDMS inserts contained holes (wells) that contained multiple microwells, allowing for cell seeding and medium exchange in multiple microwells simultaneously, minimizing differences between the microwells due to pipetting inaccuracy. The microwells presented in this study yielded homogeneous cell aggregates with controllable size, as shown for hESC- and iPSC-derived EBs as well as for U87 glioblastoma tumor spheroids.

## 2. Materials and Methods

### 2.1. CAD Design and Rapid Prototyping

SolidWorks 2021 Premium SP2.0 software was used for the CAD designs for rapid prototyping. The CAD designs were saved in the STL extension. All the designs were printed using the FormLabs 3B+ printer (FormLabs, Somerville, MA, USA) using the Dental LT Clear Biocompatible Resin (FormLabs, MA, USA). The printing was performed using the standard and the maximal resolution and using the recommended additional printing support for a total printing time of approximately six hours. The prints were washed with 100% ethanol (Avantor, VWR, Roden, The Netherlands) for 10 min and dried at 37 °C for another 10 min. Once the prints were dry, the printing support was removed carefully, leaving the final product ready to be used.

### 2.2. Scanning Electron Microscopy

The maximal resolution capacity of the printer was analyzed by means of classical SEM. To this end, the printed molds were attached using a conductive adhesive graphene tape (DAGT, Christine Groep, Vienna, Austria) to an aluminum sample holder and vacuum sputter-coated with 5 nm chrome. Subsequently, the samples were placed under the Zeiss Supra 55 Scanning Electron Microscope (Zeiss, Oberkochen, Germany) and observed using an acceleration voltage of 3 kV.

### 2.3. PDMS Casting

The PDMS Sylgard 184 kit (Dow Corning, Freeland, MI, USA) was mixed at a 10:1 elastomer-to-crosslinker weight ratio. The components were stirred vigorously for 3 min and the resulting mixture was poured into the wells of a 6-well plate. Immediately after pouring the PDMS, the 3D-printed molds were placed onto the uncured PDMS layer, ensuring full contact between the mold and the slurry. With the molds already in place, the plates were then degassed by centrifugation at 3700 rpm for 4 min using a Centrifuge 5804R (Eppendorf, Hamburg, Germany). The plates were balanced by weight prior to centrifugation. After centrifugation, the PDMS slurry appeared clear and the plates were left undisturbed at room temperature for 10 min. The PDMS was subsequently cured at 70 °C for 3 h. After curing, the plates were allowed to cool at room temperature for 30 min before the printed molds were carefully removed (Figure 1). The resulting PDMS inserts containing the microwell features were then examined by light microscopy.

In our study, a set of 12 identical 3D-printed molds was used consistently over a period of at least six months. The molds were used once or twice every two weeks depending on the experimental planning. During this time, we did not observe any impact on the quality of PDMS inserts or EB/spheroid formation.

### 2.4. PDMS Microwell Sterilization and PDMS Hydrophobicity

After casting the PDMS, the inserts were placed in a 70% ethanol bath, a sterilization method reported in multiple studies using PDMS [20,21], and sonicated for 15 min using the Trans-sonic TP690 (Elma Schmidbauer, Singen, Germany). The inserts were washed twice using DBPS and placed in 6-well plates. Alternatively, UV light or autoclaving (121 °C, 20 min) are used to sterilize the PDMS as a substrate for 3D cell aggregate formation. The hydrophobicity was measured using a custom-made water contact angle goniometer, where 5 µL of water was placed on a PDMS surface sterilized with ethanol, UV light, or autoclaving. The molds were casted and sterilized one day before or on the day of the experiment. The inner angle between the substrate and the medium drop was measured with custom-made software developed on MatLab (Mathworks Inc., Natick, MA, USA). To further study the hydrophobicity of the sterilized PDMS, the samples were incubated in mTeSR™ Plus (StemCell Technologies, Vancouver, BC, Canada) for 4, 24, 48, 72 h, and 7 days.

### 2.5. Monolayer Culture of Human Embryonic Stem Cells

Human embryonic stem cells (Hues 9) were purchased from Lonza (Basel, Switzerland). The cells were cultured using cultureware coated with Vitronectin XF™ (StemCell Technologies, Vancouver, BC, Canada). The vitronectin was thawed at room temperature and mixed gently in DPBS (Gibco, Thermo Fischer Scientific, Grand Island, NY, USA) at a final concentration of 10 µg/mL. For each well of a six-well plate, 1 mL of diluted vitronectin was used to coat the cultureware. The plates were left with vitronectin at room temperature for one hour before being used. After incubation, the vitronectin was removed and wells washed with 2 mL of DPBS. Subsequently, without letting the wells dry, the DPBS was removed and 1 mL of complete mTeSR™ Plus supplemented with 10 µM ROCK inhibitor (StemCell Technologies, Vancouver, BC, Canada) was added. The plates were left in the incubator until the cell seeding.

The cells were thawed and placed in a 10 mL tube, then 5 mL of DMEM/F12 with L-Glutamine (Gibco, Thermo Fischer Scientific, NY, USA) was added, and the cells were centrifuged at 50× *g* for 5 min, allowing only cell clumps to be in the bottom of the tube. The supernatant was discarded and the cell pellet was resuspended in 2 mL of complete mTeSR™ Plus supplemented with 10 µM ROCK inhibitor, pipetting gently for a maximum of 2 times using a P1000 pipette. Subsequently 1 mL of the cell suspension was added to each of the previously prepared vitronectin-coated wells. Ideally, small cell clumps would be floating and distributed homogeneously on the well. After 24 h, small cell colonies were observed, and the medium was exchanged with 2 mL of fresh mTeSR™ Plus supplemented with 10 µM ROCK inhibitor. After changing the medium the first time, the medium was changed every two days until the colonies reached 70% confluency.

The cells were washed twice for cell passaging with 2 mL of DPBS, then 500 µL of ReLeSR™ (StemCell Technologies, Vancouver, BC, Canada) was added to each well. The plates were incubated for 30 s at room temperature before removing the ReLeSR. Next, the wells were incubated at 37 °C for 5 min and 500 µL of mTeSR was added to each well to lift up the cell colonies. Then, the cells were pipetted twice using a P1000 pipette to break up the bigger colonies. Depending on the cell density, 50–200 µL of the cell suspension was seeded into previously prepared coated culture plates, as explained above.

### 2.6. Human Embryonic Stem Cells Embryoid Body Formation

When the colonies reached 70% confluency, the wells were washed twice using 2 mL of DPBS. After washing, 500 µL of Accutase™ (Invitrogen, Thermo Fischer Scientific, NY, USA) was added to each well and the plates were incubated at 37 °C for 5 min. Then, 1 mL of mTeSR was added to dilute the accutase and the suspension was pipetted up and down multiple times to dissociate the colonies into single cells. The cells were subsequently counted, and 2.5 × 10^3^ cells/microwell was seeded. The number of cells added per hole depended on the mold used. Before cell seeding, the PDMS inserts were preconditioned with 100 µL of mTeSR and the plates with the conditioning medium were centrifuged for one minute at 1000× *g*. The 100 µL of the cell suspension was immediately added to each insert hole, and the plates were centrifuged for 5 min at 400× *g*. After centrifugation, the plates were incubated for 24, 48, and 72 h. After optimizing the microwells’ size and shape, the inserts were compared with Aggrewell™ 800 (StemCell Technologies, Vancouver, BC, Canada) and the Primesurface^®^ (PHCbi, Etten-Leur, The Netherlands) microwell plates using the same amount of cells per microwell. For the Aggrewell plates, it is recommended to use anti-adherence rinsing solution (AARS) (StemCell Technologies, Vancouver, BC, Canada). The Primesurface^®^ and Aggrewell plates and microwell inserts were tested with and without AARS.

### 2.7. Human Induced Pluripotent Stem Cell Culture

The MSBO14 cell line was obtained from the MSiPS Biobank (MSCNN, https://www.msips-biobank.nl (accessed on 8 May 2023)). The hiPSC line was generated using non-integrative reprogramming (episomal) of peripheral mononuclear blood cells from blood samples of an MS-sibling donor. iPSC cell line was karyotyped to ensure genomic stability, and its pluripotency and morphology were assessed. hiPSCs were cultured on Matrigel (MG, Corning; cat. no. 354277, Bedford, MA, USA) coated six-well plates suitable for human cell culture in mTeSR Plus medium supplemented with Mycozap (Lonza, Basel, Switzerland, cat. no. VZA-2011) for a minimum of two passages after thawing before being used for differentiation. hiPSCs were used for experiments between passages 17 and 30. In short, MG was thawed on ice and diluted in DMEM/F12 according to manufacturer instructions and 1.5 mL of MG-DMEM/F12 coating solution was added to each cell culture plate and incubated at 37 °C for a minimum of 1 h or at 4 °C overnight. Freshly thawed hiPSC cell clusters were subsequently transferred drop-wise to a conical 15 mL Falcon tube containing 10 mL DMEM/F12 and centrifuged at 50× *g* for 5 min. Cells were carefully resuspended in 2 mL mTeSR+ containing 10 µM ROCK inhibitor and Mycozap and added to one well of a MG-coated six-well cell culture plate. The medium was changed the following day after cells had adhered. hiPSCs were passaged when the plate reached 70–80% confluence, using ReLeSR™ as described in Section 2.5.

### 2.8. hiPSC Embryoid Body Formation

hiPSCs were dissociated with Accutase, as described in Section 2.6, counted, and plated in mTeSR + supplemented with 10 µM ROCK inhibitor and Mycozap at 2.5 × 10^3^ cells/microwell or 1 × 10^6^ cells/25 mL cell culture flask. Both the PDMS microwells as well as the 25 mL cell culture flasks were pre-coated with AARS; cell culture plates containing the PDMS microwells were centrifuged at 400× *g* for 2 min to allow cells to distribute in the microwells evenly. EBs were allowed to form for four days, changing medium every other day, before EB size was assessed.

### 2.9. U87 Glioblastoma Cell Culture and 3D Spheroid Formation

Glioblastoma U87 cells were cultured in Dulbecco’s Modified Eagle medium + GlutaMAX™ (Gibco, Grand Island, NY, USA; cat. no. 31966-021) containing 10% Fetal Bovine Serum (Bodinco BV, Almar, The Netherlands; cat. No. S00QB2000B) and 1% Pen/Strep antibiotics (ThermoFisher Scientific, Grand Island, NY, USA; cat. no. 15140122) (DMEM complete medium). Cells were maintained at 37 °C in a humidified atmosphere with 5% CO_2_. After 3 days, cells were washed with PBS, detached by trypsination, and collected by centrifugation at 120× *g* for 5 min. After counting, 2.5 × 10^3^ or 5.0 × 10^3^ cells/microwell was added to each hole previously coated with AARS in a final volume of 200 µL DMEM complete medium. The plates were centrifuged at 500× *g* for 5 min to spin the cells down into the microwells. After centrifugation, the plates were incubated for 24, 48, and 72 h. Without AARS, U87 cells did not form spheroids (Appendix A).

### 2.10. Spheroid Cell Aggregate Imaging and Size Measurement

After 24, 48, and 72 h of incubation, the cell aggregates produced on the PDMS inserts and the Aggrewell and Primesurface plates with and without AARS were imaged using a standard 5× bright field microscope with a scale of 0.7733 pixel/µm. The shape and radius of the cell aggregates were analyzed using Fiji (NIH, Bethesda, MD, USA) with a semiautomatic macro. The macros code can be found in the Appendix A.

### 2.11. Statistical Analysis

Statistical analyses were performed using Graphpad Prism 9. One-way or two-way analysis of variance (ANOVA) was applied to assess mean differences between groups. The results are presented as the mean ± standard deviation (SD), where *p* ≤ 0.05 is considered significant and marked on each graph with the asterisk symbol (*). Figure legends describe the number of experimental repeats (n) and the statistical test used.

## 3. Results and Discussion

### 3.1. Microwell Design and Printing Resolution Outcome

The first CAD design consisted of 19 semi-spheres of 2 mm radius (r) (Figure 2a) as microwells. The STL files were printed under the standard and maximum resolution of the printer. The 3D-printed molds were used to cast PDMS, and the microwells were observed under a light microscope. It was found that the standard (Figure 2b) and maximum (Figure 2c) resolution have a minimal layer thickness of 52 ± SD 3 µm and 22 ± SD 2 µm, respectively, which correspond with the aspects presented by the manufacturer. The standard resolution might cause the cells to sediment on the concentric rings and might cause the formation of multiple cell aggregates. Therefore, the maximal resolution was used for further experiments. The microwells were designed with full conical (45° Slant height) and semi-spherical geometrical shapes. The shapes were truncated at 75% and 50% of the total height, and six different radii (*r*) of both shapes were tested, varying from *r* = 1.5 mm to *r* = 250 µm in steps of 250 µm (Figure 2d). The different microwells were placed on 7 mm cylinders to resemble a well of a 96-well plate (Figure 2e).

The two smallest radii (*r* = 500 µm and *r =* 250 µm) of both complete geometrical shapes were analyzed under SEM (Figure 3). The SEM showed that the *r* = 250 µm semi-spherical and conical shapes were not entirely smooth. As the geometrical shape of the microwells influences the spheroid shape during cell aggregation [4], these printing flaws might affect the spheroid morphology, which is not ideal for the formation and production of homogeneous 3D cell aggregates. Therefore, the radius *r* = 250 µm was not considered for further experiments.

### 3.2. PDMS Hydrophobicity Post-Sterilization

For cell cultures, all substrates must be sterile. Although autoclaving is the gold-standard method of lab material sterilization, this technique has a negative impact on the mechanical properties of soft materials and some polymers [22]. Other sterilization techniques, like gamma-ray irradiation or ethylene oxide (EtO), are relatively expensive. In the case of EtO, it can lead to remaining vapor toxicity that restricts its use in the biomedical industry [23]. For cell culture equipment, UV light and ethanol sterilization are basic and easy sterilization techniques that can also be applied to PDMS [21]. In the case of microwells, the cells are confined to a limited space and the environment’s mechanical properties are sensed by the cells by exerting forces on their surroundings [20]. When the cells are on a hydrophobic substrate, cell aggregation increases as cells cannot adhere to the substrate [24]. Figure 4a shows how the water contact angle of a substrate is measured. Figure 4b shows that the water contact angle of the PDMS substrates is relatively high after sterilization, considering that hydrophobic surfaces have little affinity with water and have water contact angles of >90° (hydrophobic threshold) [25]. Although the autoclaving on day 1 (105.8° ± SD 0.5°) induces significant differences compared to the UV-light (108.1° ± SD 0.3°) and the control (108.2° ± SD 1.0°), the surface remains hydrophobic, meaning that cell adhesion will be prevented, allowing for cell aggregation to occur. On day 7, the water contact angle for the UV-light-treated (106.7° ± SD 0.5) and the autoclaved (104.7° ± SD 0.8) PDMS are significantly lower as compared to the ethanol-treated control (108.3° ± SD 0.6). Despite this difference, the angle still remains more than 13° above the hydrophobic threshold. However, it is clear that the PDMS’ hydrophobicity slightly decreases after autoclaving or UV-light exposure (Figure 4b). Although there is a significant statistical difference between ethanol-sterilized and autoclaved PDMS on day 1 and between ethanol-sterilized and UV-light-treated or autoclaved PDMA on day 7 after treatment, these differences are not technically relevant as the water contact angle remains well above 90°. In this study, all the substrates for the following experiments were sterilized using 70% ethanol for ease of use. However, the possibility of sterilizing by autoclaving is important for its (future) use in clinical applications. Figure 4c shows that the hydrophobicity of the PDMS after sterilization is not affected by the cell culture medium components over time. The water contact angle measured 107.2° ± 0.3 for the control, 103.8° ± 1.1 at 4 h, 105.4° ± 2.1 at 24 h, 104.7° ± 2.3 at 48 h, 103.3 ° ± 2.5 at 72 h, and 103.1° ± 0.3 after 7 days of incubation in mTeSR stem cell medium.

### 3.3. Effect of Geometrical Shape and Size of Microwells on Embryoid Body Formation

After casting the different shapes and sizes of the test molds, cells were seeded per hole to yield 2.5 × 10^3^ or 5 × 10^3^ Hues9 embryonic stem cells per microwell. The truncated geometries and the microwells with radii bigger than 500 µm generated multiple heterogeneously-sized cell aggregates per microwell (Appendix A). Ideally, one spherical EB should be formed per microwell. Although both non-truncated geometries with *r* = 500 µm satisfied the condition of forming a single EB/microwell, ESCs in the conical microwell formed a non-spherical EB [4,9]. In contrast, ESCs in the semi-spherical non-truncated microwells always formed a single spherical EB (Appendix A). Therefore, the semi-spherical non-truncated microwells with *r* = 500 µm were used for further experiments.

### 3.4. The PDMS Microwells ENABLE Consistent Formation of Size-Controlled EBs

The microwells with *r* = 500 µm and complete semi-spherical shape were seeded with 2.5 × 10^3^ or 5 × 10^3^ Hues9 embryonic stem cells/microwell. The resulting EBs were analyzed 24, 48, and 72 h after seeding (Figure 5). For the EBs in the 2.5 × 10^3^ cells/microwell condition, the EB radius remained stable between 24 h (114.4 µm ± SD 4.0) and 48 h (115 µm ± SD 3.2), showing a significant increase only at 72 h (144.5 µm ± SD 4.2). In contrast, using 5 × 10^3^ cells/microwell produced larger EBs at all stages with significant radius increases at each time point: the EBs *r* was 152.5 µm ± SD 1.7 at 24 h, 175.5 µm ± SD 5.7 at 48 h, and 216.5 µm ± SD 3.5 at 72 h. These results demonstrate that EB size is strongly dependent on the initial number of stem cells per microwell and that EB growth over time occurs in a seeding-density-dependent manner. The lower-density condition showed minimal expansion during the first 48 h, whereas the higher-density condition exhibited continuous growth throughout the 72 h period.

As the size of the EB can have an impact on the cell lineages that will emerge upon differentiation [26], and considering that EBs of sizes between 100–300 µm have higher proliferation, viability, and differentiation potential [9], the designed and fabricated microwells using 3D printed molds for PDMS casting provide a reliable platform for controlling EB size through both initial seeding density and incubation duration. This tunability highlights the potential of the microwell system for application in developmental biology studies and tissue engineering.

### 3.5. The PDMS Microwells Enable Production of Size-Controlled EBs and Demonstrate Superior Performance Compared to ULA Suspension Culture Flasks

To assess the capacity of EB formation using iPSCs in the PDMS microwells, 2.5 × 10^3^ iPSCs/microwell were incubated for four days and compared to EB formation in an ultra-low adherent (ULA) flask. ULA flasks are commonly used for EB formation due to lower costs compared to commercial well plates for EB formation [27,28]. As seen in Figure 6a, a large number of aggregates was formed in the ULA flask. However, they have heterogeneous sizes and, more importantly, they are not consistently spherical, which is an essential characteristic of EBs if intended for differentiation [9]. Figure 6b shows that iPSCs in ULA flasks formed aggregates of multiple sizes, predominantly with radii smaller than 100 µm (75.4 ± SD 8.6) and a few aggregates with radii between 100 and 300 µm. On the contrary, the PDMS microwells generated EBs with radii of 226.8 µm ± SD 32.4, which is an optimal size range for cell differentiation [9].

### 3.6. The PDMS Microwells Enable High Production of EBs and Demonstrate Superior Performance to Commercially Available ULA Plates

Our PDMS microwells’ performance was compared with two commercially available plates (Aggrewell and Primesurface) for the formation of iPSC-derived EBs. For Aggrewell plates, the manufacturer recommends rinsing the wells with anti-adherence rinsing solution (AARS). Therefore, all plates were tested with and without the AARS. The Primesurface plates are available in multiple geometrical shapes, of which the rounded (U) and the conical bottom shapes (V) were tested; 2500 cells/microwell was used for every application to compare the microwells’ performance. Figure 7 shows that the Primesurface plates generated cell aggregates at least two times bigger than those produced with our PDMS microwells. Specifically, the radius was *r =* 256.8 µm ± SD 27.1, 310.1 µm ± SD 31.5, and 437.1 µm ± SD 22.0 at 24, 48, and 72 h, respectively, for the rounded bottom plate without AARS, and *r =* 246 µm ± SD 23.6 at 24 h, 302.5 µm ± SD 27.5 at 48 h, and 422.5 µm ± SD 25.1 at 72 h for the same plates with AARS. As can be seen in Figure 7, the radii of the EBs formed in the conical Primesurface plates are not significantly different from those in the round bottom wells, i.e., 269.6 µm ± SD 19.6, 326.6 µm ± SD 23.4, and 470.7 µm ± SD 16.5, at 24, 48 and 72 h, respectively, for the conical bottom plate without AARS and 250 µm ± SD 12.2 at 24 h, 318.7 µm ± SD 23.6 at 48 h, and 440.6 µm ± SD 15.6 at 72 h with AARS. In the Primesurface plates—with wells of 7 mm diameter—the cell aggregates do not experience contact inhibition [29], which leads to the formation of bigger spheroids compared to in the PDMS microwells. However, in these bigger EBs the differentiation capacity and the viability of the inner cell mass can be negatively affected [9,26]. As shown in Appendix A, in our microwells with *r* > 500 µm, multiple cell aggregates of different sizes were formed instead of one EB, whereas on the Primesurface plates, single EBs were formed. This likely happened because of the printing resolution of our microwell molds, which caused cells to sediment onto the different print layers upon centrifugation.

In addition, the commercial Aggrewell™ 800 plates were compared with the PDMS microwells. These plates have a square pyramid base length of 800 µm, which is comparable to the *r* = 500 µm of the PDMS-designed microwells. In Figure 8, it can be seen that at 24h and 48 h after cell seeding, the AARS pretreatment did not significantly affect the formation of EBs. However, it is remarkable that at 48 h, the SD of the EBs’ size on the Aggrewell plates (124.8 µm ± SD 10.2 and 120.8 µm ± SD 11.7 with and without AARS) is at least two times larger than the EBs formed on the PDMS microwells (*r* = 119.4 µm ± SD 5.5 without AARS and 128.1 µm ± SD 4.4 with AARS). In addition, the size of the EBs formed on the Aggrewell plates almost doubles from 24 h to 48 h of incubation. This can be explained by the small interspace between truncated pyramid microwells on the Aggrewell plates allowing unintended cell/aggregate overflow between adjacent wells, which leads to the presence of multiple EBs in a single well and possibly their fusion, resulting in EBs of a bigger size (Figure 8 and Appendix A). On the Aggrewell 800 plates, the number of single EBs formed remained below 50% due to cell overflow (Appendix A). On the contrary, the PDMS microwells consistently showed one EB per microwell, indicating a better EB yield. Furthermore, the EBs showed consistent size and shape, which is ideal for a uniform nutrient supply and optimal cell mechanics in EBs [9].

Moreover, the PDMS inserts can be produced in-house on-demand and used separately depending on the needs. Additionally, the molds can be printed and cast for 6-, 12-, 24-, and 48-well plates. Even more important, the medium in the PDMS inserts can be replaced without disturbing the cell aggregates, which for example opens the door to EB differentiation without disturbing cell–substrate mechanics.

### 3.7. PDMS Microwells Enable High Production of Tumor Spheroids from Adherent U87 Glioblastoma Cells

Next, the inserts were tested for spheroid formation of glioblastoma cells (U87) to extend the capacity for consistent spheroid formation in the PDMS microwells from cells that grow in suspension and spontaneously from spheroids, like EBs, to adherent cell lines. Figure 9 shows successful formation of U87 cell spheroids in the PDMS microwells following AARS coating. By using different cell concentrations, it was possible to control the size of the spheroids at 24, 48, and 72 h of incubation. Following the plating of 2500 cells/microwell, tumor spheroids were formed with *r =* 115.8 µm ± SD 2.5, 123.4 µm ± SD 2.5, and 134.8 µm ± SD 3.6 at 24, 48, and 72 h, respectively. For 5000 cells/microwell, the tumor spheroids showed *r =* 142.0 µm ± SD 1.2, 148.4 µm ± SD 0.5, and 152.8 µm ± SD 1.2. To ensure optimal spheroid formation by U87 cells, the use of AARS was necessary (cf. Figure 9 and Appendix A). The formation of size-controlled tumor cell spheroids is crucial for controlling oxygen gradients [30], mesenchymal transitions linked to cancer cell migration [31,32], evaluating the effects of drug treatments [33], and exploring interactions between tumors and the immune system [34].

## 4. Conclusions

This study presents a 3D-printed mold manufactured by vat photopolymerization for the production of PDMS microwells for high-yield generation of homogeneous cell aggregates. Microwells were designed and fabricated with full conical and semi-spherical geometrical shapes or truncated at 75% and 50% of the total height and with six different radii (250 µm, 500 µm, 750 µm, 1 mm, 1.25 mm, and 1.5 mm). The 500 µm radius semi-sphere microwell was shown to maximize homogeneity of embryoid bodies and tumor spheroids and was evaluated in a direct comparison with commercially available systems, i.e., ULA flasks, Primesurface^®^ plates, and Aggrewell plates.

Our PDMS microwells showed excellent control over spheroid size and shape, in contrast to spheroids cultured in ULA flasks that suffered from size heterogeneity and non-sphericity. Compared to spheroids cultured in Aggrewell plates, the PDMS microwells showed a better EB yield, while both devices require AARS to generate spheroids from adherent cell lines. Another advantage of the presented PDMS microwells is the significantly reduced cost: the 3D resin for creating the 3D-printed molds and the PDMS for the inserts are relatively cheap compared to the high prices of the commercially available microwell and low-attachment plates. Another key feature of the presented system is that the manufacturing process of PDMS casting, i.e., soft lithography, is straightforward and uses equipment that is commonly available in biology labs. Furthermore, the possibility of customizing the mold’s design is a critical feature that allows for adaptation to the needs of individual researchers. Of note, one set of 3D-printed molds was used consistently throughout the study, with microwells fabricated fresh for each experiment. Mold re-use for up to six months did not affect PDMS casting or microwell quality, which demonstrates the robustness and practical usability of the system for routine lab workflows. Moreover, regular printer maintenance can ensure consistent mold production, supporting reproducible and reliable results. In summary, the semi-spherical PDMS microwells with *r* = 500 µm were validated as a platform for fast, simple, cost-effective high-throughput production of homogeneous spheroids, specifically EBs (formed from ESC and iPSCs) and U87 tumor spheroids.

Recent studies that have reported on 3D printed molds and PDMS casting for microwell generation to induce formation of 3D cell aggregates, typically presented cone-shaped or pyramidal microwells [34,35,36,37], but not semi-spherical geometries. However, one other study that compared different microwell geometries, including truncated cone, semi-sphere, and pyramid, for blastocyst generation for IVF similarly reported best performance for the semi-sphere microwells [38], possibly because of their better resemblance to the curved surface of cell aggregates.

Next experiments will focus on the biological validation beyond spheroid morphology, which will depend on the cell type used as well as on the specific research question that is being explored. In this regard, potential limitations of the use of PDMS in cell culture systems should be taken into account, particularly its tendency to absorb hydrophobic compounds, which, for example, may influence effective drug concentrations in drug screening applications.

## Figures and Tables

**Figure 1 pharmaceutics-18-00056-f001:**
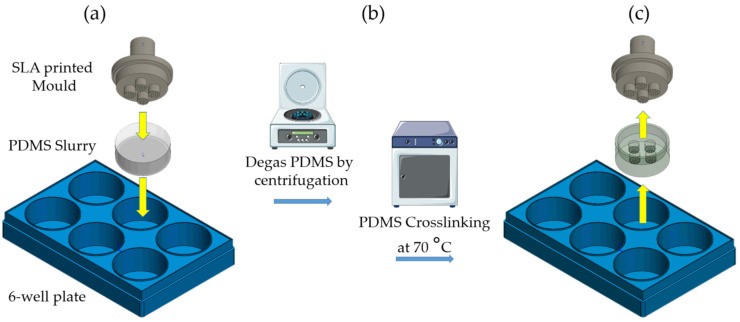
Microwell formation using 3D printed molds on PDMS. (**a**) PDMS imprinting, (**b**) degasification and crosslinking, and (**c**) insert retrieval. The figure was partly generated using Servier Medical Art, provided by Servier, licensed under a Creative Commons Attribution 3.0 unported license.

**Figure 2 pharmaceutics-18-00056-f002:**
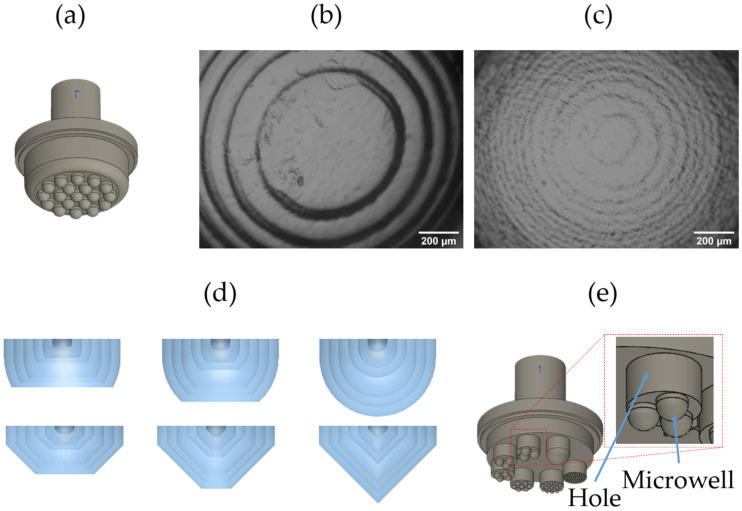
Microwell optimization design. (**a**) CAD’s first microwells design, (**b**) standard printing resolution 52 ± 3 µm, (**c**) maximal printing resolution 22 ± 2 µm of the PDMS microwells, and (**d**) conical and semi-spherical CAD designs for microwells of *r* = 250, 500, 750 µm,1 mm, 1.25 mm, and 1.5mm. (**e**) CAD design with all the microwell radii for a 6-well plate configuration.

**Figure 3 pharmaceutics-18-00056-f003:**
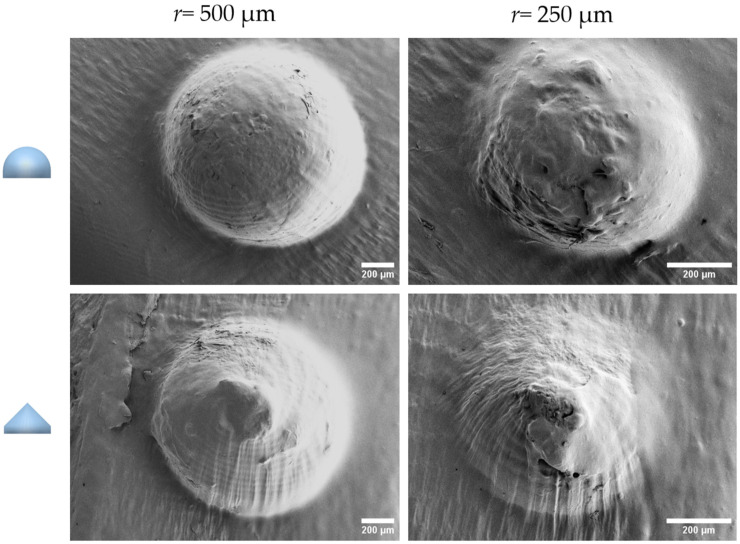
Microwell mold analysis by scanning electron microscopy (SEM). SEM of *r* = 500 µm and *r* = 250 µm microwell molds in conical and semi-spherical shapes using the maximum VP printing resolution.

**Figure 4 pharmaceutics-18-00056-f004:**
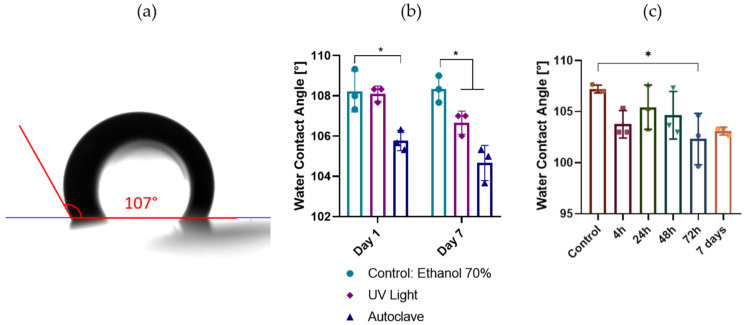
PDMS hydrophobicity properties after sterilization. (**a**) Water contact angle measurement by the custom-made water contact angle goniometer. (**b**) Water contact angle of PDMS sterilized with ethanol 70% (control), UV-light, and autoclaving (n = 3, repeated measures two-way ANOVA Dunnet test * *p* ≤ 0.05). (**c**) Water contact angle of PDMS incubated in mTeSR plus where the control is a sample that is not incubated in mTeSR (n = 3, ordinary one-way ANOVA Dunnets test * *p* ≤ 0.05). All data are presented as mean ± SD.

**Figure 5 pharmaceutics-18-00056-f005:**
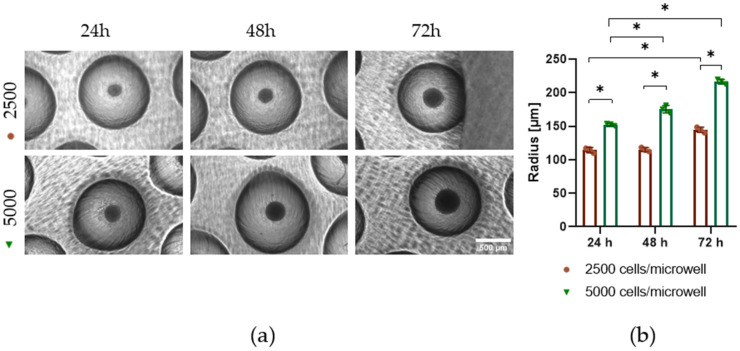
Cell seeding density controls EB size. (**a**) Light microscopy of EBs formed from Hues9 ESCs seeded at 2.5 × 10^3^ or 5 × 10^3^ cells/microwell at 24, 48, and 72 h after seeding. (**b**) n = 3, repeated measures ANOVA Sidak multiple test comparison * *p* ≤ 0.05.

**Figure 6 pharmaceutics-18-00056-f006:**
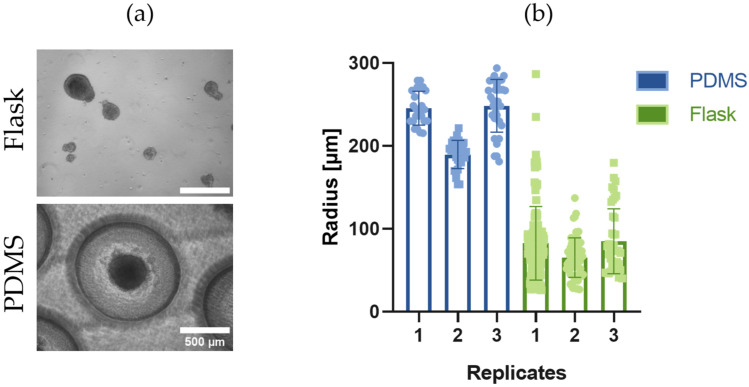
The PDMS microwells perform superiorly to ULA culture flasks in EB formation. (**a**) Light microscopy of iPSCs-EBs after four days of incubation in ULA flasks or PDMS microwells. (**b**) Individual and average values for EB radii in 3 independent experimental replicates using the PDMS microwells (min. value *r* = 153.5 µm, max. value = 293.8 µm) and ULA flasks (min. value *r* = 25.8 µm, max. value = 286.7 µm). (n = 3, unpaired *t*-test with Welch’s correction).

**Figure 7 pharmaceutics-18-00056-f007:**
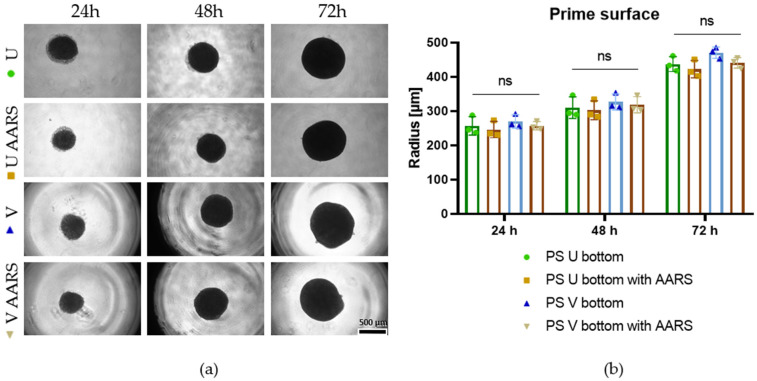
Spherical EBs on Primesurface^®^ plates. (**a**) Light microscopy of iPSC-derived EBs formed on conical (V) and round bottom (U) Primesurface^®^ plates with and without anti-adherence rinsing solution (AARS) at 24, 48, and 72 h after seeding. The wells of both U- and V-shaped plates have a diameter of 7.5 mm. The U-shaped wells give the appearance of floating EBs, while the V-shaped wells reflect light in a manner that produces a dark rim around the edges. Scalebar = 500 µm (right down corner). (**b**) Graph showing average EB radii under different conditions (n = 3, repeated measures ANOVA Tukey’s multiple test comparison ns *p* ≥ 0.05).

**Figure 8 pharmaceutics-18-00056-f008:**
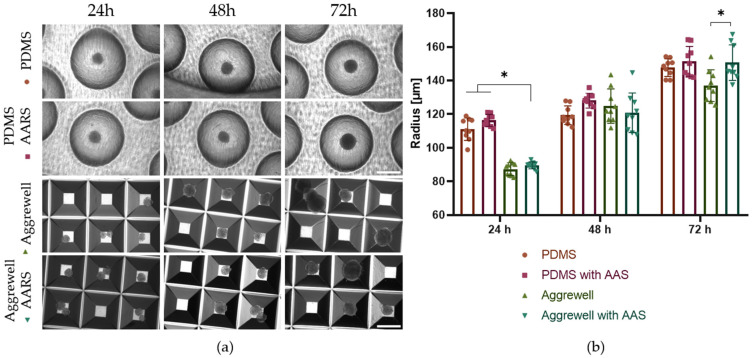
Comparison of EB formation in PDMS microwells and Aggrewell plates. (**a**) Light microscopy of EBs on the semi-spherical PDMS microwells (*r* = 500 µm) and EBs on Aggrewell^®^ plates with and without anti-adherence rinsing solution at 24, 48, and 72 h after seeding. Scalebar = 500 µm. (**b**) Graph represents individual and average values for EB radii for the different conditions (n = 9, repeated measures ANOVA Dunnet’s multiple test comparison, * *p* ≤ 0.05).

**Figure 9 pharmaceutics-18-00056-f009:**
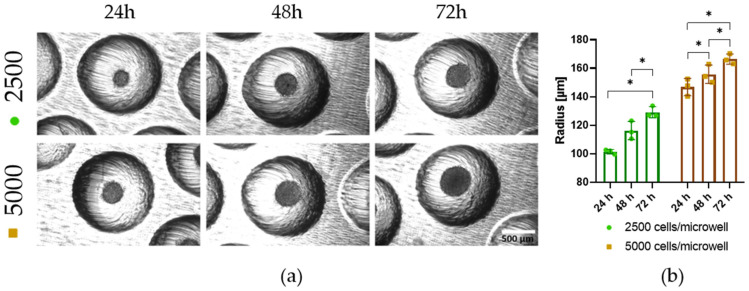
U87 tumor spheroid formation in PDMS microwells. (**a**) Light microscopy of tumor spheroids with 2500 and 5000 cells/microwell at 24, 48, and 72 h after AARS coating and seeding. (**b**) n = 3, repeated measures ANOVA Sidak’s multiple test comparison * *p* ≤ 0.05.

## Data Availability

All the data supporting the findings of this study are available from the corresponding author upon request.

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
