# Peer review of "Development of a Microwell System for Reproducible Formation of Homogeneous Cell Spheroids"

_pharmaceutics, 2025, doi:10.3390/pharmaceutics18010056_

Round 1

Reviewer 1 Report

Comments and Suggestions for Authors

Page 2 : Correct the sentence « With this technique, typically multiple aggregates are formed within one drop and their sizes and shapes are inconsistent [14]. »

Materials and methods : 2.3 Casting: It is not clear at which step of the protocol is the PDMS degasified, and when are the printed molds applied on PDMS.

2.4 The autoclave conditions should be explained (temperature and humidity).

Figure 4.C. The y axis is not correctly spelled.

Section 3.4 : The conclusion should be revised, as the results discussed in this section show that the EBs size depends on the concentration of stem cells per microwell, and, for some concentrations, on the time.

Why do the EBs size presented in Figure 7 have a higher dispersity than the results presented in Figure 6 at 72h for the same cell concentration.

The title of Section 3.5 should be revised since the results do not show the possibility of high production of EBs.

Author Response

1) Page 2 : Correct the sentence “With this technique, typically multiple aggregates are formed within one drop and their sizes and shapes are inconsistent [14].”

Answer: The sentence has been corrected (p.3 lines 76-78).

2) Materials and methods : 2.3 Casting: It is not clear at which step of the protocol is the PDMS degasified, and when are the printed molds applied on PDMS.

Answer: We apologize for the lack of clarity. Section 2.3 PDMS casting has been rewritten to improve clarity (p.5 lines 138-149).

3) 2.4 The autoclave conditions should be explained (temperature and humidity).

Answer: We added the relevant information regarding the autoclave program in section 2.4. The program lasts 20 min at 121 ºC. Sterilization by pressurized saturated steam is expected to have very high humidity, essentially 100%.

4) Figure 4.C. The y axis is not correctly spelled.

Answer: We thank the reviewer for noting the error and have corrected it.

5) Section 3.4 : The conclusion should be revised, as the results discussed in this section show that the EBs size depends on the concentration of stem cells per microwell, and, for some concentrations, on the time.

Answer: We agree that the conclusion should explicitly state that EB size is influenced both by the number of stem cells seeded per microwell and, at certain concentrations, by the incubation time. As our results show, microwells seeded with 2.5×10³ cells maintain a similar EB size between 24 and 48 h, with a significant increase only at 72 h, whereas microwells seeded with 5×10³ cells display a continuous and statistically significant increase in EB size over all time points measured. These observations confirm that EB growth dynamics are not only cell-number dependent, but also time-dependent, and that the magnitude of the time effect varies with initial seeding density. To reflect these findings more directly, we have revised section 3.4 and its conclusion as follows (p.12/13 lines 252-368):

“The microwells with r= 500 µm and complete semi-spherical shape were seeded with 2.5x103 or 5x103  Hues9 embryonic stem cells/microwell. The resulting EBs were analyzed 24, 48, and 72 hours after seeding (Figure 5). For the EBs in the 2.5x103  cells/ microwell condition, the EB radius remained stable between 24h (114.4 µm ± SD 4.0) and 48h (115 µm ± SD 3.2), showing a significant increase only at 72h (144.5 µm ± SD 4.2). In contrast, using  5x103  cells/microwell produced larger EBs at all stages with significant radius increases at each time point: the EBs r was 152.5 µm ± SD 1.7 at 24h, 175.5 µm ± SD 5.7 at 48 h, and 216.5 µm ± SD 3.5 at 72h. These results demonstrate that EB size is strongly dependent on the initial number of stem cells per microwell, and that EB growth over time occurs in a seeding density-dependent manner. The lower-density condition showed minimal expansion during the first 48 hours, whereas the higher-density condition exhibited continuous growth throughout the 72-hour period.

As the size of the EB can have an impact on the cell lineages that will emerge upon differentiation [26], and considering that EBs of sizes between 100-300 µm have higher proliferation, viability, and differentiation potential [9], the designed and fabricated microwells using 3D printed molds for PDMS casting provide a reliable platform for controlling EB size through both initial seeding density and incubation duration. This tunability highlights the potential of the microwell system for application in developmental biology studies and tissue engineering.”

6) Why do the EBs size presented in Figure 7 have a higher dispersity than the results presented in Figure 6 at 72h for the same cell concentration.

Answer: The higher dispersity in EB size observed in Figure 7 compared with Figure 6 (in the revised manuscript Figures 6 and 5), despite using the same initial cell concentration, can be explained by the use of different cell types and their intrinsic biological differences. Specifically, the EBs shown in Figure 6 were generated using Hues9 ESCs, while the EBs in Figure 7 were generated from iPSCs. Although both lines are pluripotent, iPSCs are widely reported to exhibit greater heterogeneity in colony morphology, proliferation rate, and cell–cell adhesion, partly due to variability introduced during reprogramming and residual epigenetic memory. These differences translate into more heterogeneous aggregation dynamics, which has been shown in previous studies to produce EBs with larger size variability compared with hESC-derived EBs under identical microwell conditions (doi: 10.1073/pnas.0910012107;  doi: 10.1371/journal.pgen.1004432). Therefore, the increased spread in EB diameters in Figure 7 is consistent with the expected biological behavior of iPSCs compared to hESCs.

In Figure 7, each point corresponds to an individual EB radius, making the full intrinsic variability directly visible.

7) The title of Section 3.5 should be revised since the results do not show the possibility of high production of EBs.

Answer: The reviewer is correct and the title has been adjusted accordingly. It now reads: The PDMS microwells enable production of size-controlled EBs and demonstrate superior performance compared to ULA suspension culture flasks

Reviewer 2 Report

Comments and Suggestions for Authors

This manuscript addresses an important and timely challenge in 3D cell culture: the need for reliable, accessible, and cost-effective methods to generate homogeneous cell aggregates. The authors present a custom 3D-printed microwell stamp system designed to produce PDMS-based semi-spherical and conical microwells for forming pluripotent stem cell aggregates and tumor spheroids. The work is well motivated, as current commercial platforms and laboratory techniques often suffer from variability, high cost, or operational complexity. By integrating high-resolution vat-photopolymerized molds with PDMS casting, the authors propose a versatile and customizable alternative that appears capable of improving aggregate uniformity while reducing fabrication and operational expenses. The study is clearly positioned within the growing demand for scalable 3D culture technologies and therefore has potential relevance to researchers working in developmental biology, disease modeling, and drug screening. However, several aspects still require clarifications.

  • Originality of Concept

The manuscript introduces a practical and cost-effective approach to generating homogeneous 3D cell aggregates using custom 3D-printed microwell molds cast in PDMS. While microwell-based spheroid culture is not conceptually new, the authors’ integration of vat-photopolymerization printing, customizable CAD-based mold design and soft lithography represents a meaningful methodological refinement. The novelty lies in improving accessibility, reducing cost, and enhancing reproducibility of spheroid culture systems—an area of clear importance across stem cell and cancer research.

  • Experimental Design
  1. The authors present systematic optimization of microwell geometry (shape, truncation, radius) leading to rational selection of a 500 µm full semi-sphere as the optimal configuration. Furthermore, a broad investigation was performed against multiple commercial systems, which strengthens the study’s translational relevance. Nevertheless, Effects of mold re-use or inter-batch variability in printing/casting are not described.
  2. The manuscript demonstrates a versatile use of the platform to generate aggregates consisted of ESCs, iPSCs, and U87 glioblastoma cells. However, Biological validation beyond morphology (e.g., viability, differentiation potential, early gene expression) would further strengthen the conclusions.
  3. The authors conducted appropriate characterization of PDMS surface hydrophobicity following sterilization and exposure to culture media. Yet, PDMS absorption of small hydrophobic molecules is not discussed, although relevant for drug screening applications.
  • Results Display and Clarity of Presentation
  1. Overall, the microscopy images presented in the manuscript are generally clear and appropriately annotated. However, a specific concern might raise regarding the figure 7, which presents a comparison between ULA-derived aggregates and PDMS-derived aggregates. The comparison is not fully rigorous due to imaging inconsistencies. The ULA brightfield image lacks a scale bar, and the legend does not provide magnification information. As a result, the true size and morphology of the aggregates cannot be directly compared to the PDMS-derived aggregates. Additionally, the aggregates appear to be imaged in different contexts (free-floating vs. confined microwells), which may bias the visual interpretation.
  2. The manuscript currently includes 10 figures, many of which contain multiple panels. While each figure conveys relevant information, the overall number is relatively high for a methods-oriented study and may overwhelm readers. Several figures present incremental findings that could be combined or streamlined without loss of scientific clarity.

Author Response

This manuscript addresses an important and timely challenge in 3D cell culture: the need for reliable, accessible, and cost-effective methods to generate homogeneous cell aggregates. … By integrating high-resolution vat-photopolymerized molds with PDMS casting, the authors propose a versatile and customizable alternative that appears capable of improving aggregate uniformity while reducing fabrication and operational expenses. … However, several aspects still require clarifications.

Answer: We thank the reviewer for noting the timely nature of our manuscript.

Experimental Design

1) The authors present systematic optimization of microwell geometry (shape, truncation, radius) leading to rational selection of a 500 µm full semi-sphere as the optimal configuration. Furthermore, a broad investigation was performed against multiple commercial systems, which strengthens the study’s translational relevance. Nevertheless, Effects of mold re-use or inter-batch variability in printing/casting are not described.

Answer: We thank the reviewer for the question, as this is useful information to add. The following information was added to the paper under Materials and Methods section 2.3 (p.5 lines 150-153) and discussed in the conclusion section:  In our study, a set of 12 identical 3D-printed molds was used consistently over a period of at least six months. The molds were used once or twice every two weeks depending on the experimental planning. During this time, we did not observe any impact on the quality of the PDMS inserts or EB/spheroid formation.

2) The manuscript demonstrates a versatile use of the platform to generate aggregates consisted of ESCs, iPSCs, and U87 glioblastoma cells. However, Biological validation beyond morphology (e.g., viability, differentiation potential, early gene expression) would further strengthen the conclusions.

Answer: We fully agree with the reviewer that additional biological validation would further strengthen the conclusions. However, the biological validation will depend on the cell type used as well as on the specific research question that is being addressed. Preliminary experiments on EB differentiation were performed, in which EBs formed in the PDMS microwells were differentiated toward NPCs and compared to EBs formed in low-adherence flasks. In both conditions, PAX6/Nestin-positive NPCs were obtained, with no major differences in differentiation efficacy. In addition, the PDMS microwells were used to generate glioblastoma spheroids, showing high cell viability and metabolic activity, for testing the penetration of nanoparticles. These results support the compatibility of the platform with functional assays, but they fall beyond the scope of the present manuscript, which focuses on the development and characterization of the platform up to the level of reproducibly generating size-controlled spheroids.

3) The authors conducted appropriate characterization of PDMS surface hydrophobicity following sterilization and exposure to culture media. Yet, PDMS absorption of small hydrophobic molecules is not discussed, although relevant for drug screening applications.

Answer: The reviewer is correct and this is a limitation of the use of PDMS, already known from the microfluidics field. We now discuss this limitation in the Conclusions section (p. 19 lines 516-520): Next experiments will focus on the biological validation beyond spheroid morphology, which will depend on the cell type used as well as on the specific research question that is being explored. In this regard, potential limitations of the use of PDMS in cell culture systems should be taken into account, particularly its tendency to absorb hydrophobic compounds, which for example may influence effective drug concentrations in drug screening applications.

Results Display and Clarity of Presentation

4) Overall, the microscopy images presented in the manuscript are generally clear and appropriately annotated. However, a specific concern might raise regarding the figure 7, which presents a comparison between ULA-derived aggregates and PDMS-derived aggregates. The comparison is not fully rigorous due to imaging inconsistencies. The ULA brightfield image lacks a scale bar, and the legend does not provide magnification information. As a result, the true size and morphology of the aggregates cannot be directly compared to the PDMS-derived aggregates. Additionally, the aggregates appear to be imaged in different contexts (free-floating vs. confined microwells), which may bias the visual interpretation.

Answer: We apologize for the perceived inconsistency in Figure 7 (Figure 6 in the revised manuscript). The brightfield images of the iPSC-EBs in the ULA flasks and the PDMS microwells were made on the same microscope with the same settings (including magnification, exposure, and brightness parameters). We have now added a size bar to both images for clarification.

5) The manuscript currently includes 10 figures, many of which contain multiple panels. While each figure conveys relevant information, the overall number is relatively high for a methods-oriented study and may overwhelm readers. Several figures present incremental findings that could be combined or streamlined without loss of scientific clarity.

Answer: We understand the point raised by the reviewer. To focus on the comparisons between our PDMS-based system and commercially available materials to generate spheroids, we have moved Figure 5 (that shows the selection of the best microwell design for formation of single spherical spheroids) to the supplementary figures. 

Reviewer 3 Report

Comments and Suggestions for Authors

This manuscript develops a simple 3D-printed mold + PDMS microwell system to generate homogeneous EBs and tumor spheroids. Overall, the idea is good, the experiments are clear, and the data show real improvement compared to some commercial microwell plates. The paper is generally well written. But several points need to be addressed before I can recommend acceptance.

  1. It seems that the resolution of the printed part in Figure 3 is not very high. Would it be possible to improve the printing resolution by varying the printing parameters or printing direction of the designed parts? Or this limitation is caused by the machine available to the author.
  2. There are some papers that reported using 3D-printed molds + PDMS for microwells before. It is recommended that the authors state more clearly what is new/the difference here.
  3. Reusability and long-term stability are not discussed. It is recommended that the author provide some discussion or experiment or reference to show that the PDMS molds prepared by this method can be used multiple times.

Author Response

1) It seems that the resolution of the printed part in Figure 3 is not very high. Would it be possible to improve the printing resolution by varying the printing parameters or printing direction of the designed parts? Or this limitation is caused by the machine available to the author.

Answer: The maximum resolution achievable with our 3D printer is 22 ± 2 µm when printing solid parts at a 45° angle. In Figure 3, the mold for the 250 µm microwells shows noticeable quality degradation, which is consistent with the physical limits of the printer. Attempting to print features below this size results in uneven surfaces, which would translate into the PDMS casting and lead to irregular embryoid body (EB) shapes or even allow EBs to move out of the wells. Therefore, the observed resolution limitation is inherent to the printer and the feature size, rather than the printing parameters or orientation alone.

2) There are some papers that reported using 3D-printed molds + PDMS for microwells before. It is recommended that the authors state more clearly what is new/the difference here.

Answer: The reviewer is correct. Indeed, some studies have employed 3D‑printed molds, PDMS casting and microwell arrays for spheroid/multicellular‑aggregate formation.
However, prior reports often show proof-of-principle formation of spheroids (with hydrogel or PDMS microwells), but they rarely provide a direct, quantitative comparison across multiple commercial systems under identical conditions (e.g., ULA flasks, Aggrewell plates). In contrast, our study includes a head-to-head comparison against commercial platforms, demonstrating superior yield, size uniformity, and cost-efficiency of our in-house developed microwells.

Furthermore, most studies present one microwell design, typically cone-shaped or pyramidal, for one specific application. In contrast, we first systematically optimized microwell geometry (shape, truncation, radius) and selected a 500 µm radius semi-sphere, a geometry not reported in those  studies, to maximize homogeneity of iPSC‑derived EBs and tumor spheroids. Another study that did compare different geometries (truncated cone, semisphere and pyramid) for blastocyst generation for IVF, similarly reported best performance of the semisphere microwells. In our revised manuscript we have now included these comparisons with prior work (plus references) in the Conclusions section (p.19 lines 509-515): Recent studies that have reported on 3D printed molds and PDMS casting for microwell generation to induce formation of 3D cell aggregates, typically presented cone-shaped or pyramidal microwells [34-37], but not semispherical geometries. However, one other study that compared different microwell geometries, including truncated cone, semisphere and pyramid,  for blastocyst generation for IVF, similarly reported best performance for the semisphere microwells [38], possibly because of its better resemblance to the curved surface of cell aggregates.

3) Reusability and long-term stability are not discussed. It is recommended that the author provide some discussion or experiment or reference to show that the PDMS molds prepared by this method can be used multiple times.

Answer: The following information was added to the paper under Materials and Methods section 2.3 and discussed in the conclusion section:

In our study, a set of 12 identical 3D-printed molds was used consistently over a period of at least six months. The molds were used once or twice every two weeks depending on the experiment planning. During this time, we did not observe any impact on the quality of the PDMS inserts or EB/spheroid formation. In fact, our optimized design (with 9 times 19 microwells) is being used for over 2 years now with just one of the microwell shapes showing some degradation, leading to one unsuitable microwell out of 171.

Round 2

Reviewer 2 Report

Comments and Suggestions for Authors

After a thorough re-evaluation of the authors' responses to my initial review, it is evident that they have adequately addressed the critical questions raised and made the necessary revisions to the manuscript. These revisions significantly enhance the clarity and impact of the manuscript, providing a more complete and cohesive presentation of the research.

Therefore, I recommend accepting the manuscript in its most recent version.